

# Acute effects of anterior thigh foam rolling on hip angle, knee angle, and rectus femoris length in the modified Thomas test

Andrew D. Vigotsky[1], Gregory J. Lehman[2], Bret Contreras[3], Chris Beardsley[4], Bryan Chung[5] and Erin H. Feser[1]

[1] Kinesiology Program, School of Nutrition and Health Promotion, College of Health Solutions, Arizona State University, Phoenix, AZ, USA
[2] Private Practice, Toronto, Ontario, Canada
[3] School of Sport and Recreation, Auckland University of Technology, Auckland, New Zealand
[4] Strength and Conditioning Research Limited, London, UK
[5] Hand Program, University Health Network, Toronto, Ontario, Canada

Corresponding author
Andrew D. Vigotsky,
avigotsky@gmail.com

## ABSTRACT

**Background.** Foam rolling has been shown to acutely increase range of motion (ROM) during knee flexion and hip flexion with the experimenter applying an external force, yet no study to date has measured hip extensibility as a result of foam rolling with controlled knee flexion and hip extension moments. The purpose of this study was to investigate the acute effects of foam rolling on hip extension, knee flexion, and rectus femoris length during the modified Thomas test.

**Methods.** Twenty-three healthy participants (male = 7; female = 16; age = 22 ± 3.3 years; height = 170 ± 9.18 cm; mass = 67.7 ± 14.9 kg) performed two, one-minute bouts of foam rolling applied to the anterior thigh. Hip extension and knee flexion were measured via motion capture before and after the foam rolling intervention, from which rectus femoris length was calculated.

**Results.** Although the increase in hip extension (change = +1.86° (+0.11, +3.61); $z(22) = 2.08$; $p = 0.0372$; Pearson's $r = 0.43$ (0.02, 0.72)) was not due to chance alone, it cannot be said that the observed changes in knee flexion (change = −1.39° (−5.53, +2.75); $t(22) = −0.70$; $p = 0.4933$; Cohen's $d = −0.15$ (−0.58, 0.29)) or rectus femoris length (change = −0.005 (−0.013, +0.003); $t(22) = −1.30$; $p = 0.2070$; Cohen's $d = −0.27$ (−0.70, 0.16)) were not due to chance alone.

**Conclusions.** Although a small change in hip extension was observed, no changes in knee flexion or rectus femoris length were observed. From these data, it appears unlikely that foam rolling applied to the anterior thigh will improve passive hip extension and knee flexion ROM, especially if performed in combination with a dynamic stretching protocol.

## INTRODUCTION

Foam rolling (FR) is a ubiquitous intervention, performed by athletes during both preparation for and following physical activity. FR is postulated to be a form of self-myofascial release, despite no research investigating whether FR directly influences fascia. Therefore, it is perhaps presumptuous to refer to the fascia in its name, let alone as its mediator. For the purposes of this study, FR and similar modalities, such as massage sticks, will be referred to as self-manual therapy. Previous self-manual therapy and FR work has been shown to increase range of motion (ROM) (*Behara & Jacobson, 2015*; *Halperin et al., 2014*; *Jay et al., 2014*; *MacDonald et al., 2013*; *Schroeder & Best, 2015*; *Škarabot, Beardsley & Stirn, 2015*; *Sullivan et al., 2013*), attenuate delayed onset muscle soreness (DOMS) (*Macdonald et al., 2014*; *Pearcey et al., 2015*), reduce arterial stiffness (*Okamoto, Masuhara & Ikuta, 2014*), and improve vascular endothelial function (*Okamoto, Masuhara & Ikuta, 2014*). Importantly, FR has also been shown not to have detrimental effects on physical performance (*Behara & Jacobson, 2015*; *Halperin et al., 2014*; *Healey et al., 2014*; *MacDonald et al., 2013*; *Sullivan et al., 2013*).

Quadriceps injuries are a commonplace in sport (*Orchard & Seward, 2002*), and the rectus femoris is the most commonly injured quadriceps muscle (*Cross et al., 2003*; *Speer, Lohnes & Garrett, 1992*). Risk factors for injury appear to be multifactorial (*Mendiguchia et al., 2013*). Of particular interest are hip flexor strength and flexibility, as these can be modified through training and warm-up (*Mendiguchia et al., 2013*). Should FR be an efficacious methodology for increasing rectus femoris extensibility, it may allow athletes to train through a greater ROM. Training through a greater ROM would allow athletes to not only see greater gains in strength throughout a greater ROM, but also increase their total ROM (*Hartmann et al., 2012*; *McMahon et al., 2014*; *Morton et al., 2011*; *Wyon, Smith & Koutedakis, 2013*). Because both strength and flexibility are risk factors for rectus femoris strain injury, doing so may reduce risk of injury (*Mendiguchia et al., 2013*).

The effectiveness of self-manual therapy in increasing ROM is of particular interest, as flexibility appears to be a risk factor for muscle strain injury (*Mendiguchia et al., 2013*). As noted by *Schroeder & Best (2015)*, despite the heterogeneity of previous studies, FR does appear to be an efficacious intervention for increasing flexibility. Only a couple of studies have investigated the effects of self-manual therapy on hip flexor—namely, rectus femoris—extensibility, as measured by either knee flexion or hip extension. The first to do so was *MacDonald et al. (2013)*, who found that two, one-minute bouts of FR applied to the anterior thigh increased knee flexion ROM by 10° and 8° at 2 and 10 min post-intervention, respectively. More recently, *Bushell, Dawson & Webster (2015)* investigated the effects of three, one-minute bouts of FR applied to the anterior thigh on hip extension angle during a dynamic lunge. The intervention was completed once per week for three weeks. The only increase in hip extension noted was during the second week; no changes were found in the first or third weeks. Investigators did report a slight increase in hip extension from the first to second week (3.7° vs. 0.34° (control)), but it cannot be said that this increase was not due to chance alone. The effects of FR on static measures of hip extension, knee flexion, and consequently, rectus femoris length have not

yet been investigated. Therefore, the purpose of this study was to investigate the effects of two, one-minute bouts of FR of the anterior thigh on acute hip extension ROM, knee flexion ROM, and rectus femoris length. It was hypothesized that FR of the anterior thigh will acutely increase hip extension ROM, knee flexion ROM, and rectus femoris length.

## METHODS

### Study design

This study used a within-subject repeated measures design. Data was collected during one experimental session. Hip extension and knee flexion were measured both before and after one bout of FR, which was carried out by FR the hip flexor muscle group on the anterior thigh.

### Participants

As per an *a priori* power analysis ($\alpha = 0.05$; $\beta = 0.80$; expected difference = 2.95°; Cohen's $d = 0.54$) for an increase in hip extension using G*Power (*Faul et al., 2007*), 23 participants (male = 7; female = 16) were recruited from a student population via flyers placed on a University campus and presented to Kinesiology and Exercise and Wellness classes. Participants would have been excluded only if they currently had a back or lower extremity musculoskeletal or neuromuscular injury or pain, but no participants reported such an injury. Before each participant was scheduled for testing, the participants were asked about their current injury status. Participants were provided a verbal explanation of the study, and read and signed an Informed Consent and Physical Activity Readiness Questionnaire (PAR-Q) before beginning. Any participant that would have answered "Yes" to any of the questions on the PAR-Q would have been excluded, but none did. Participants' age, height, and weight were then measured. The study was approved by the Institutional Review Board at Arizona State University (IRB ID: STUDY00001660).

### Procedures

A ten minute standardized warm up procedure followed. This warm up consisted of five minutes on an Airdyne bike, two sets of 20 body weight squats, two sets of 10 leg swings in both the frontal and sagittal planes, and two sets of 10 body weight lunges.

Once the aforementioned 10-minute warm-up was completed, reflective markers were adhered to participants' skin or tight fitting garments on the lateral femoral epicondyle, greater trochanter, lateral malleolus, and iliac crest, halfway between the PSIS and ASIS and spaced 10 cm apart. These methods differ slightly from those presented by *Kuo, Tully & Galea (2008)*, as the PSIS and ASIS markers were placed closer to the midaxillary line so they would not be blocked from the camera by the table. Once placed, the markers were not removed until after the final (post-FR) testing procedure. Should the participant's tight fitting garment have had any potential marker distractions (e.g., reflective logos), they were covered with masking tape.

Participants then performed one FR intervention, utilizing a 91.44 (L) × 15.24 (D) cm polypropylene foam roller (Perform Better, West Warwick, RI) directed at the right anterior thigh for two, 60-second bouts. While lying prone, participants were instructed

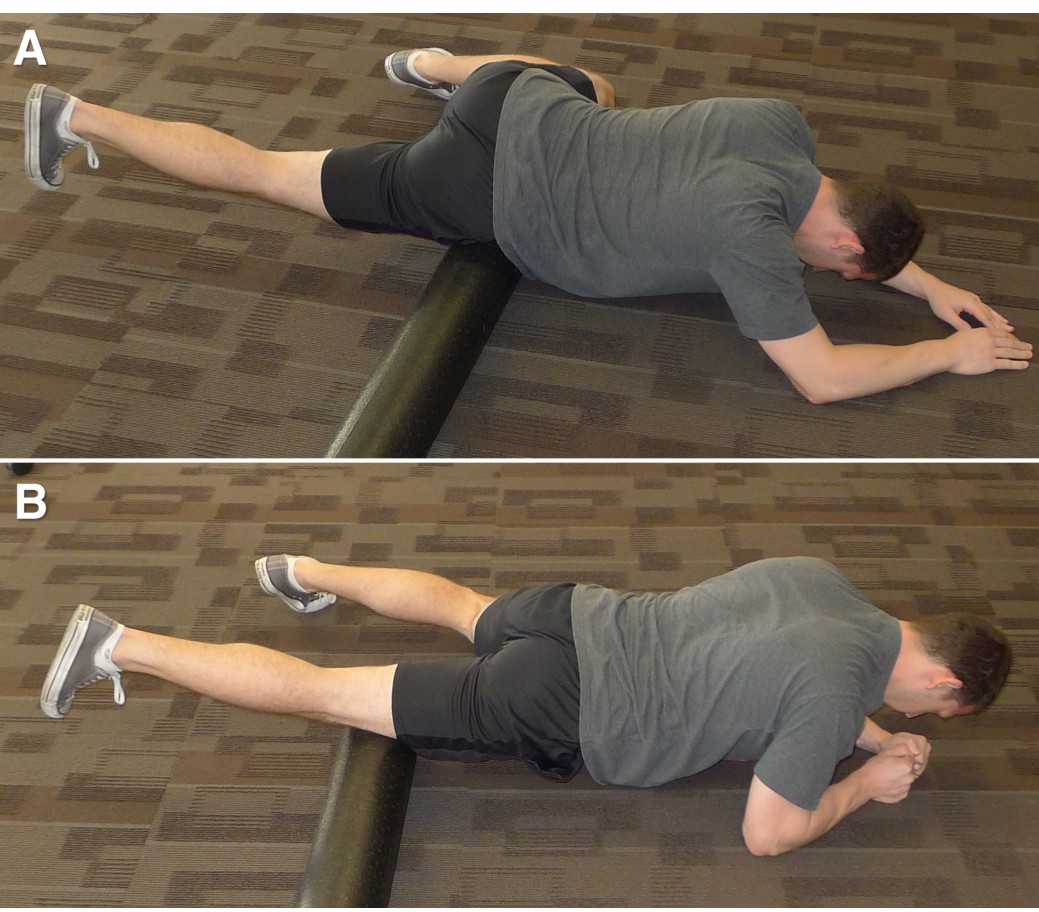

**Figure 1** Starting (A) and ending (B) position of the foam roll protocol.

to place their body weight on the foam roller, starting at the proximal aspect of the thigh (just inferior to the ASIS) and rolling down the thigh in a kneading-like fashion, slowly reaching the knee (Fig. 1). Once the foam roller reached the superior knee, participants were instructed to return the roller to the starting position and continue the sequence for the remainder of the 60 s (*MacDonald et al., 2013*). Participants were instructed to complete the intervention at a slow pace (seconds/repetition). Following a thirty-second break, the participant repeated this intervention.

Within one minute of completing the second bout of FR, participants' hip extension and knee flexion ROM were re-tested.

## Testing and analysis

Hip extension and knee flexion were measured as the participant performed the modified Thomas test. Hip extension values were calculated by subtracting the four-point angles that the four markers create from 90°. Knee flexion values were calculated by subtracting the three-point angles that the three markers create from 180°. Two-dimensional sagittal plane motion capture were obtained using a 120 Hz camera, set to 30 Hz (Basler Scout scA640-120; Basler Vision Technologies, USA), and motion analysis software (MaxTRAQ

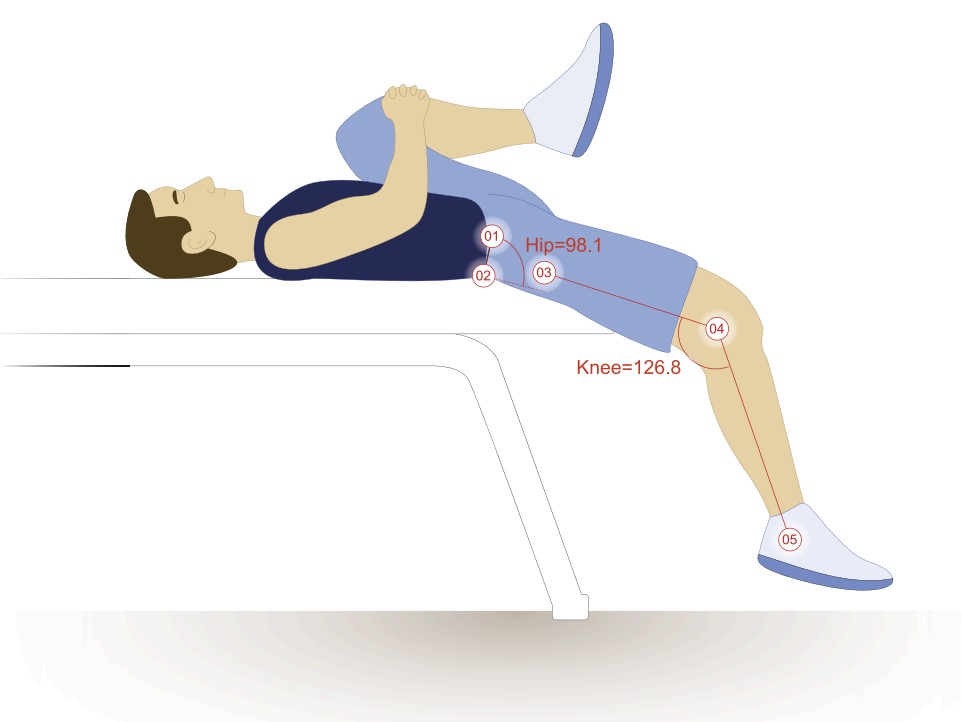

**Figure 2 Hip and knee joint calculations.** The illustrated participant would have a hip extension angle of 8.1° (98.1° − 90°) and a knee flexion angle of 53.2° (180° − 126.8°). Illustration credit: Ji Sung Kim

2D; Innovision Systems Inc., Columbiaville, Michigan, USA). Marker digitization was also completed in MaxTRAQ, using auto-digitization and auto-tracking, as to prevent investigator bias. These methods (motion capture) differ substantially from those previously described (*Harvey, 1998*) in that the hip angle was measured relative to the pelvis rather than the plinth (Fig. 2). This prevented lumbopelvic movement from confounding the results of the modified Thomas test, which can severely impact the test's reliability (*Kim & Ha, 2015*). Furthermore, measuring these angles via motion capture presumably allows for more reliable and objective measures, as the same points are being utilized to calculate the angle with each measurement trial. Doing so has demonstrated very high levels of reliability for knee flexion (ICC = 0.98; SEM = 1.0°) (*Peeler & Leiter, 2013*) and hip extension (ICC = 0.90–0.95; SEM = 2.0°) (*Wakefield et al., 2015*). The average of three tests was used for each participant's reported measure. Rectus femoris length was estimated using the regression equations and coefficients provided by *Hawkins & Hull (1989)*, and is presented relative to length at neutral (that is, hip and knee at 0°) (Eq. (1)); for example, 1.020 would represent a 2% increase from resting length, or 102% of resting length. Similar methods and presentation of data were utilized by *Thelen et al. (2004)* and *Vigotsky et al. (2015)*.

$$l_{RF} = \frac{1.107 - (1.50 \cdot 10^{-3})\theta_{hip} + (1.99 \cdot 10^{-3})\theta_{knee}}{1.107} \tag{1}$$

**Table 1** Descriptive statistics of participants.

| Sex | n | Age (years) | Height (cm) | Body mass (kg) |
|---|---|---|---|---|
| Male | 7 | 21.00 ± 1.63 | 179.54 ± 6.90 | 83.24 ± 11.25 |
| Female | 16 | 22.06 ± 3.84 | 165.74 ± 6.53 | 60.91 ± 10.64 |
| Total | 23 | 22.00 ± 3.30 | 169.95 ± 9.18 | 67.71 ± 14.90 |

The averages of the three pre- and post-FR measures of hip extension ROM, knee flexion ROM, and calculated rectus femoris length were entered into Stata 13 (StataCorp LP, College Town, Texas, USA). Shapiro–Wilk tests were performed to ensure normality. For normal data, paired samples $t$-tests were performed. Any data found to be non-parametric were compared using Wilcoxon paired-samples signed-rank tests. Alpha was set to 0.05. Parametric effect sizes (ES) were calculated by Cohen's $d$ using the formula $d = \frac{M_d}{s_d}$, where $M_d$ is mean difference and $s_d$ is the standard deviation of differences (*Becker, 1988*; *Morris, 2008*; *Smith & Beretvas, 2009*). This method is slightly different than the traditional method of calculating Cohen's $d$, as it calculates the within-subject ES rather than group or between-subject ES. Cohen's $d$ was defined as small, medium, and large for 0.20, 0.50, and 0.80, respectively (*Cohen, 1988*). Non-parametric ES were reported in terms of Pearson's $r$. Pearson's $r$ was defined as small, medium, and large for 0.10, 0.30, and 0.50, respectively (*Cohen, 1988*). Confidence limits of 95% (95% CL) for ES were also calculated. Because a small number of preplanned comparisons were made, no correction was employed.

## RESULTS

Twenty-three healthy participants (Table 1) were recruited and underwent two, 1-minute bouts of FR the anterior thigh. Measures of hip extension did not meet parametric assumptions, but knee flexion and rectus femoris length did. Although the increase in hip extension (change $= +1.86°$ $(+0.11, +3.61)$; $z(22) = 2.08$; $p = 0.0372$; Pearson's $r = 0.43$ $(0.02, 0.72)$) was not due to chance alone, it cannot be said that the observed changes in knee flexion (change $= -1.39°$ $(-5.53, +2.75)$; $t(22) = -0.70$; $p = 0.4933$; Cohen's $d = -0.15$ $(-0.58, 0.29)$) or rectus femoris length (change $= -0.005$ $(-0.013, +0.003)$; $t(22) = -1.30$; $p = 0.2070$; Cohen's $d = -0.27$ $(-0.70, 0.16)$) were not due to chance alone (Table 2).

## DISCUSSION

The purpose of this study was to determine if FR applied to the anterior thigh increases hip extension, knee flexion, and rectus femoris length during the modified Thomas test. It appears that the moderate to large effect observed for hip extension was not due to chance alone. However, it cannot be said for certain that this increase was not at the expense of a decrease in knee flexion, especially since no real change in rectus femoris length was observed. Although prior research would consider the observed increase in hip extension to be clinically relevant—as the observed 45.24% increase in hip extension exceeds the 10% threshold (*Roach & Miles, 1991*)—one cannot truly compute a relative change of joint kinematics, as joint angles are an interval scale, and not ratio scale (*O'Donoghue,*

**Table 2 Individual and mean (±SD) changes in hip extension ROM, knee flexion ROM, and calculated rectus femoris length pre- and post-FR.**

|  | Sex | Hip extension (°) | | | Knee flexion (°) | | | Rectus femoris length | | |
|---|---|---|---|---|---|---|---|---|---|---|
|  |  | Pre- | Post- | Δ | Pre- | Post- | Δ | Pre- | Post- | Δ |
| 1 | M | 17.0 | 16.5 | −0.5 | 40.9 | 62.5 | 21.6 | 1.051 | 1.090 | 0.039 |
| 2 | F | −4.5 | −6.7 | −2.1 | 48.1 | 58.9 | 10.8 | 1.093 | 1.115 | 0.022 |
| 3 | F | 8.9 | 10.1 | 1.2 | 53.5 | 62.7 | 9.2 | 1.084 | 1.099 | 0.015 |
| 4 | M | 4.0 | 4.5 | 0.4 | 49.4 | 57.8 | 8.4 | 1.083 | 1.098 | 0.015 |
| 5 | F | −1.1 | 0.9 | 2.0 | 24.8 | 33.2 | 8.4 | 1.046 | 1.059 | 0.012 |
| 6 | M | 20.5 | 18.6 | −1.8 | 47.6 | 51.1 | 3.5 | 1.058 | 1.067 | 0.009 |
| 7 | F | −3.6 | −1.7 | 1.9 | 53.4 | 59.5 | 6.1 | 1.101 | 1.109 | 0.008 |
| 8 | F | 16.3 | 17.2 | 0.9 | 50.6 | 53.1 | 2.5 | 1.069 | 1.072 | 0.003 |
| 9 | F | 10.2 | 10.4 | 0.2 | 53.4 | 55.2 | 1.8 | 1.082 | 1.085 | 0.003 |
| 10 | F | 12.1 | 14.9 | 2.7 | 55.9 | 59.2 | 3.3 | 1.084 | 1.086 | 0.002 |
| 11 | M | −5.5 | −6.1 | −0.6 | 49.1 | 48.0 | −1.1 | 1.096 | 1.094 | −0.001 |
| 12 | F | −5.3 | −9.8 | −4.5 | 64.7 | 57.0 | −7.7 | 1.124 | 1.116 | −0.008 |
| 13 | F | 10.8 | 13.0 | 2.2 | 58.6 | 54.5 | −4.1 | 1.091 | 1.080 | −0.010 |
| 14 | F | 2.7 | 1.8 | −0.9 | 49.9 | 43.8 | −6.1 | 1.086 | 1.076 | −0.010 |
| 15 | M | −3.1 | 4.0 | 7.2 | 49.1 | 48.5 | −0.6 | 1.093 | 1.082 | −0.011 |
| 16 | F | 5.2 | 8.6 | 3.4 | 53.6 | 45.2 | −8.4 | 1.089 | 1.070 | −0.020 |
| 17 | F | 2.7 | 5.0 | 2.3 | 58.3 | 47.4 | −11.0 | 1.101 | 1.078 | −0.023 |
| 18 | F | 5.3 | 5.8 | 0.5 | 57.6 | 45.1 | −12.5 | 1.096 | 1.073 | −0.023 |
| 19 | M | −4.5 | 4.8 | 9.4 | 39.2 | 32.4 | −6.8 | 1.077 | 1.052 | −0.025 |
| 20 | F | 3.6 | 1.2 | −2.5 | 55.0 | 38.8 | −16.2 | 1.094 | 1.068 | −0.026 |
| 21 | F | 9.2 | 13.8 | 4.6 | 59.2 | 46.9 | −12.3 | 1.094 | 1.066 | −0.028 |
| 22 | M | −3.5 | 10.6 | 14.0 | 55.6 | 49.9 | −5.7 | 1.105 | 1.075 | −0.029 |
| 23 | F | −0.6 | 2.2 | 2.9 | 56.0 | 40.8 | −15.2 | 1.102 | 1.070 | −0.031 |
| $\bar{x}$ |  | 4.2 ± 7.8 | 6.1 ± 7.8 | 1.9 ± 4.0 | 51.5 ± 8.2 | 50.1 ± 8.7 | −1.4 ± 9.6 | 1.087 ± 0.018 | 1.082 ± 0.017 | −0.005 ± 0.019 |

*2015*). Additionally, and perhaps more importantly, the increase in hip extension did not exceed the previously-reported SEM of 2.0° (*Wakefield et al., 2015*), implying that the mean change observed in this trial would not be clinically detectable within or between individuals.

Of interest is the inter-individual variability to the FR intervention, as there were responders, non-responders, and even decreases in rectus femoris length observed in individuals (Table 2). For example, participant #1's rectus femoris length increased by 3.9%, while participant #23's rectus femoris length decreased by 3.1%, and participant #11 nearly did not experience any change (−0.1%). Furthermore, those who had similar changes in rectus femoris length did not necessarily experience those changes from the same place; participant #22 experienced a large increase in hip extension with a decrease in knee flexion, while participant #23 experienced a small increase in hip extension and a large decrease in knee flexion, but both participants experienced similar decreases in rectus femoris length (−0.029 and −0.031, respectively). Interestingly, there did not appear to be differences in responses between genders (Table 2). These results differ slightly from *MacDonald et al. (2013)*, who found an increase in knee flexion with the hip fixed in

extension. However, it should be noted that MacDonald and colleagues' ROM testing utilized a less objective protocol because the participants' passive range of movement was measured while the experimenter actively applied a force to flex their knee.

Our protocol involved two 1-minute bouts FR on the anterior thigh, which was identical to the dosage investigated by both *MacDonald et al. (2013)* and *Markovic (2015)*, who each assessed changes in knee flexion angle following FR of the anterior thigh and reported statistically significant increases. However, it was smaller than the dosage assessed by *Bushell, Dawson & Webster (2015)*, who investigated changes in hip extension angle during a lunge movement following three 1-minute bouts of FR on the anterior thigh and reported statistically significant increases. Given that our protocol involved dosages at the lower end of what has previously been utilized in the literature, it is possible that it could potentially have been insufficient to bring about changes in flexibility. However, since similar protocols have reported increases in flexibility and since no previous trial has yet identified a dose–response effect following self-manual therapy (*Bradbury-Squires et al., 2015*; *Sullivan et al., 2013*), this would seem unlikely. Furthermore, several other trials making use of smaller dosages of FR, albeit in other muscle groups, have all reported statistically significant increases in flexibility (*Halperin et al., 2014*; *Škarabot, Beardsley & Stirn, 2015*; *Sullivan et al., 2013*). Together, these factors suggest that the dosage used in our protocol was likely sufficient.

The only external force applied to the thigh, resulting in a hip extension moment, was the weight of the participants' lower extremity. Also, since the setup for the modified Thomas test is nearly identical each time, the moment arm about each segment (thigh and leg) is likely similar on each setup. However, it is possible that these moment arms change depending on the compliancy of the rectus femoris. The constant external force and presumably moment arm, and thus external hip extension moment, may provide insight into the mechanisms of self-manual therapy. If FR's effectiveness is a result of an increase in tissue extensibility, an increase in muscle–tendon unit length (with a related decrease in tissue stiffness) with the same applied moment would have been observed due to a shift in the length-tension curve (*Weppler & Magnusson, 2010*). However, a similar applied moment alone was not enough to elicit observable changes in rectus femoris length. Following these outcomes, it is proposed that self-manual therapy may work through an increase in stretch tolerance rather than an increase in tissue length, as an increase in tissue length or decreased stiffness would have resulted in increased rectus femoris length with the same applied tension. This is certainly possible, as potential mechanisms for manual therapy have been described to be primarily neurophysiological in nature (*Bialosky et al., 2009*). Recently, *Eriksson Crommert et al. (2015)* described similar effects after a seven-minute massage; a decrease in muscle stiffness, measured via elastography, was observed immediately following intervention, but there was no observed effect at three minutes. These findings are similar to ours, in that no changes in stiffness were observed shortly following intervention. This warrants further research utilizing dynamometry, elastography, or similar methods to measure passive joint or muscle stiffness before and following a FR protocol.

The mechanism by which FR of the anterior thigh increases flexibility at the hip or knee might inform an understanding of similar interventions intended to reduce rectus femoris strain injury risk. Reviewing rectus femoris strain injury in soccer, *Mendiguchia et al. (2013)* suggested that the reduced capacity for using the stretch-shortening cycle during the kicking action that might follow from less hip extension ROM, resulting from less hip flexor extensibility, could be key for an increased risk of strain injury. Reduced capacity for using the stretch-shortening cycle could require the rectus femoris to produce more muscle force for each kicking action, thereby increasing the rate of fatigue and consequently the risk of injury. It has been suggested that increasing muscle compliance could reduce muscle strain injury risk in the stretch-shortening cycle in general by enhancing the ability of the muscle–tendon unit to store energy (*Witvrouw et al., 2004*). Similarly, it has been argued that since strain injuries occur in stretch (*Mendiguchia et al., 2013*), a stiffer, less flexible muscle might be less likely to incur a strain injury than a compliant one (*Gleim & McHugh, 1997*; *Noonan & Garrett, 1999*; *Safran, Seaber & Garrett, 1989*). Since the findings of our investigation indicate that FR might exert its effects through an increase in stretch tolerance rather than biomechanical mechanisms (as no change in muscle length was observed and the test did not require additional tension) an increase in flexibility following FR may not provide the purported benefits that could reduce rectus femoris strain injury risk through increases in muscle compliance.

Several limitations should be noted in relation to this study. First, of worthy mention is the warm-up participants endured during this study, which was assumed to more closely mimic the warm-up that athletes would typically undergo. The warm-up protocol employed was longer than that of *MacDonald et al. (2013)*, who only had participants perform five minutes on a cycle ergometer. It is possible that the participants in our study had already maximized the potential acute rectus femoris extensibility gains before testing began (*O'Sullivan, Murray & Sainsbury, 2009*), especially since it has been shown that FR and dynamic stretching may elicit similar gains in hip flexion ROM (*Behara & Jacobson, 2015*). Therefore, the extensive warm-up protocol must be taken into account when interpreting these results. Second, the pace at which participants completed the FR intervention was not recorded, and this might have had an effect on individual outcomes (i.e., pace-dependent outcomes). Thirdly, the only external force applied to the thigh, resulting in a hip extension moment, was the weight of the participants' lower extremity. Therefore, it is difficult to form conclusions as to whether FR of the anterior thigh would allow a patient or athlete to move through a greater ROM, as the external moment of force during exercise may allow the athlete to increase his or her ROM.

## CONCLUSIONS

Although a small change in hip extension was observed, no changes in knee flexion or rectus femoris length were observed. From these data, it appears unlikely that FR applied to the anterior thigh will improve passive hip extension and knee flexion ROM, especially if performed in combination with a dynamic stretching protocol.

### Funding

The authors received no funding for this work.

### Competing Interests

The authors declare there are no competing interests.

### Author Contributions

- Andrew D. Vigotsky conceived and designed the experiments, performed the experiments, analyzed the data, contributed reagents/materials/analysis tools, wrote the paper, prepared figures and/or tables, reviewed drafts of the paper.
- Gregory J. Lehman conceived and designed the experiments, reviewed drafts of the paper.
- Bret Contreras conceived and designed the experiments, contributed reagents/materials/analysis tools, reviewed drafts of the paper.
- Chris Beardsley wrote the paper, reviewed drafts of the paper.
- Bryan Chung analyzed the data, contributed reagents/materials/analysis tools, reviewed drafts of the paper.
- Erin H. Feser conceived and designed the experiments, performed the experiments, contributed reagents/materials/analysis tools, reviewed drafts of the paper.

### Human Ethics

The following information was supplied relating to ethical approvals (i.e., approving body and any reference numbers):

Arizona State University Institutional Review Board
STUDY00001660.

### Supplemental Information

Supplemental information for this article can be found online at http://dx.doi.org/10.7717/peerj.1281#supplemental-information.

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
