# Peer review of "Acute effects of anterior thigh foam rolling on hip angle, knee angle, and rectus femoris length in the modified Thomas test"

_PeerJ, doi:10.7717/peerj.1281_

## Round 0.1 · original submission · Major Revisions

I have now received three reviews of your manuscript. The reviewers agree that is an interesting paper worth publishing, but it requires more work. The three reviewers had different questions that need to be addressed. In particular they asked for more information in relation to the experimental design and validity of the findings. I would like to see the following points addressed in the ms: sex differences; size effect when reporting the pre-post changes in RoM and muscle length; validity of the Modified Thomas Test as a measure of the Knee-RoM; the no-independence of the three measures considered; differences in dosing the foam rolling intervention compared to other literature; details of the a priori power analysis. Please add information on the reliability and measurement error of the measurement technique performed and justify your use of parametric tests. Please incorporate the suggested literature into your revision.

Apart from these points, please address the other more minor suggestions from the reviewers in your revision / response letter.

·

Basic reporting

Methods: Perhaps the authors could add subtitles throughout the methods section to help guide readers who are looking for specific sections.

Rectus Femoris muscle length: Can the authors please include some measurement units with all reports of muscle length? Currently, there were no units throughout.

Line 193: Should "knee flexion" be substituted for "hip flexion"?

Figure 1: It is unclear how the knee RoM was calculated to yield 53.2 degrees. Could the authors please describe the method of determining knee RoM more thoroughly (if they decide to omit knee RoM, this comment can be ignored)?

Experimental design

Lines 98-99: It may be beneficial for readers if the author included a more substantial description of the estimation of muscle length along with the reference.

Participants: Females have been known to have increased flexibility (especially at the hip joint) compared to males. Since the participants were predominantly female, is there a possibility that their increased flexibility may have masked effects that could have been seen in males? Thus, is it possible that the authors could touch on sex differences in the MS? This can be done in the discussion and, perhaps, the authors could further report any sex differences that were seen in the baseline measures and in pre-post FR differences.

Statistics: Could the authors please include effect sizes when reporting the pre-post changes in RoM and muscle length?

Validity of the findings

Modified Thomas Test to measure knee flexion RoM: The validity of this measure may be in question. The Thomas Test does not measure knee flexion range of motion since the shank simply hangs passively. Further, it may be more appropriate to focus the interpretation of outcomes from the Modified Thomas Test on the Hip extension. If the authors believe that the Thomas Test is a valid measure of Knee RoM, could they please include some supporting literature in the introduction. If the knee RoM is omitted from the MS as an outcome from the modified Thomas test, the discussion portion of the MS should then focus on how differences of Hip (current experiment) could differ from knee RoM in previous literature.

Interpretation of findings: If a muscle is more compliant (i.e. increased stretch tolerance), could that decrease the risk of injury from stretch? Currently, the conclusion suggests that previous work supporting FR creates a more compliant muscle, whereas the current study negates the possibility that FR increases muscle length and, thus, will not reduce the risk of injury in proceeding play. Perhaps a short section delineating the mechanism of common injuries that would be prevented by an increased muscle length and not increased muscle compliance could support the conclusion how it is currently written, otherwise, it may be appropriate to suggest that FR can increase muscle compliance but not length and, thus, could potentially reduce the risk of injury due to sudden stretch of the muscle.

Additional comments

Title: Perhaps a revision of the title would better depict the findings of the current experiment. Since the Modified Thomas Test is quite well known, could the title be changed to "Acute effects of anterior thigh foam rolling on the Modified Thomas Test and rectus femoris length"? Furthermore, throughout the abstract could the authors refer to the effects of FR on the modified Thomas test rather than hip extension and knee flexion?

·

Basic reporting

The Introduction needs to have a bit more depth related to ROM improvements following foam rolling. Side issues of changes in physical performance, or injury rates are only of tangential interest to this manuscript, and can be greatly de-emphasized or eliminated. Areas to emphasize include specific muscles or muscle groups targeted, and dosing information across the previous studies.

There are two 2015 studies available epub ahead of print that deal with foam rolling and hip motion.1,2 The authors should include these studies in the next draft of this manuscript.

The second sentence is controversial and opinionated. Why not just call it foam rolling? I recommend eliminating the second sentence and re-writing the first.

Figure 1 is of poor quality and will need to be re-done. Illustration of the 8.1 degree hip angle and 53.2 knee angle is not clear.

1. Behara B, Jacobson BH. THE ACUTE EFFECTS OF DEEP TISSUE FOAM ROLLING AND DYNAMIC STRETCHING ON MUSCULAR STRENGTH, POWER, AND FLEXIBILITY IN DIVISION I LINEMEN. Journal of orthopaedic trauma. Jun 24 2015.
2. Bushell JE, Dawson SM, Webster MM. Clinical Relevance of Foam Rolling on Hip Extension Angle in a Functional Lunge Position. Journal of strength and conditioning research / National Strength & Conditioning Association. Feb 14 2015.

Experimental design

The description of the methods are mostly clear. I especially appreciated the a priori power analysis.

The hip extension ROM and Knee flexion ROM are mislabeled as they are not ranges of motion - they are joint positions while in the Thomas Test position. Additionally, the three measures are being treated as if they are independent, when in fact they are not. As the study clearly concerns rectus femoris length (which of course depends of hip and knee position), I suggest dropping the separate analyses of hip and knee joint positions and keep the rectus femoris length analysis.

Validity of the findings

There is no information on the reliability and measurement error of the measurement technique performed.

Please also indicate if the markers were removed prior to rolling, then replaced for the post-test.

Rectus femoris length is computed and presented relative to its computed length at 0 degrees hip and knee flexion. The resulting change scores are of a very small magnitude. The joint position data is presented to a tenth of a degree (without measurement error we don't know how precise this measure really is...), but the rectus femoris data are carried out three decimal places which seems excessive. I'll let the Editor weigh in on this, but I believe the general rule is that the result of a calculation cannot be more precise than the least precise element of the calculation.

Additional comments

Elimination of hip and knee position analysis would take care of problems in the initial Discussion where the results are focused on hip extension.

Minor point but even though the external force is constant, the moment arm, therefore the moment would change with the hip in more or less extension. Also, gravity is an external load, perhaps you meant no additional external load?

Differences in dosing the foam rolling intervention compared to other literature needs to be addressed.

The conclusions are too broad and reach too far from the data of the study. This study did not measure injury risk, nor stretch tolerance, so no conclusions about these things can be made.

·

Basic reporting

The article is generally well written with appropriate paragraph and sentence structure. The introduction provides a relevant background to the topic of foam rolling with a reasonable rationale offered. I would prefer if the authors could use sub-headings throughout the Materials and Methods section. If possible the authors need to provide a higher resolution image for figure 1 - currently due to the resolution it is not clear whether the participants foot is elevated off the floor as it should be during the Modified Thomas Test. Lastly the discussion is well written and the authors should be praised for their reporting and discussion of individual participant data - would it be possible for table 2 to have participant's ordered based upon magnitude of change in your primary outcome variable (I assume in this case this is hip extension ROM considering your power analysis - if not this needs clarifying). This would allow readers to better identify groups of responders, non-responders and negative responders.

Experimental design

The authors report having conducted an a priori power analysis yet do not provide details of how this was conducted (e.g. calculations, software utilised etc.). Further theynote this was based upon an expected difference of 2.95 degrees for hip extension ROM - it needs to be made clear where this value came from or how it was derived (also to my understanding it is normal to base power analyses instead upon effect size estimates).

The authors note inclusion and exclusion criteria including exclusion based upon answering 'Yes' to any PAR-Q questions - can they report how many participants were indeed excluded and the reasons why?

For 2D kinematics the authors note the participants had markers attached to either their skin or 'tight fitting garments' - can they provide comment on whether they were aware of any marker distractions that might occur from the type of garments their participants wore?

Lines 95-97 - The authors note that use of 2D motion capture is 'presumably' more reliable and objective. If there is not any published reliability data for the method utilised here (different as you note from those described by Harvey) then it would be good to include some reliability data for the digitisation process. On this note, I assume the same investigator conducted the digitisation each time? This needs to be made clear.

Description of the intervention is well detailed, however, it could also include the 'rep duration' for the FR (i.e. how long it took for participants to move from top to bottom and vice-versa)?

Validity of the findings

Regarding the data analysis the authors need to justify their use of parametric tests by reporting results of tests for assumptions of normality (e.g. Kolomogorov-Smirnov or Shapiro-Wilk). Also, though I agree with the authors use of multiple comparison corrections here due to the related nature of the three outcomes examined, the authors need to provide explanation and justification for their choice of using this procedure (i.e. why correction was necessary/desirable and why the Bonferonni procedure was used in lieu of alternatives? The authors state they calculated 95% confidence intervals for outcomes. I assume this is what is reported in the brackets after the mean in the results section - this needs to be made clear. Also, though effect sizes using cohens d are easily calculated by the reader from table 2 it would be useful to include them. Finally (and appreciably this is perhaps a bit picky), it is more commonly accepted to report t test results as for example (t(22) = 2.20, p < 0.0383).

Additional comments

Abstract:

Need to note that participants were both male and female.

Please reword the results to something along the lines of "Based upon the Bonferonni-adjusted alpha level (p = 0.0167), there were no significant changes from pre- to post-intervention for..."

This applies to the results section in the main article also.

Introduction:

Lines 35-38 - sentence beginning "For the purposes of..." is cumbersome and requires rewording/breaking up.

Lines 50-52 - Need to provide citation to support suggestion that increasing hip extension ROM may reduce rectus femoris strain.

Materials and Methods:

Line 66 - Please indicate that 23 male and female participants were recruited and indicate number of males and females.

Line 68 - replace 'he or she' with 'they'

Line 69-70 "...participants were not excluded..." this line needs rewording - also can you indicate how many participants had prior injuries?

Results:

See above - reword.

Discussion:

Line 149 - units need to be included - this applies throughout the manuscript so please check all instances.

---

## Round 0.2 · Minor Revisions

I appreciate your carefully consideration of our reviewers suggestions. I would like you to make your conclusions without including bold statements such as those related to the rectus femoris extensibility.

·

Basic reporting

Although this MS shows that the modified Thomas Test has unchanged hip and knee angles, it may be more appropriate to say that it is unlikely that FR applied to the anterior thigh will improve "passive hip and knee ROM", since the modified Thomas Test is not a direct measure of the extensibility of the rectus femoris.

All other edits meet my satisfaction.

Experimental design

No comments

Validity of the findings

No comments.

Additional comments

Good work on the revisions. I commend you on an interesting report about foam rolling.

---

## Round 0.3 · accepted · Accept

Thank you for taking into consideration the reviewers suggestions. I am happy to accept your revision.

A note: Please, be watchful. The document includes tracked sentences that should not be there, for example in the title and methods.